# Clinical Predictors of Lung-Function Decline in Systemic-Sclerosis-Associated Interstitial Lung Disease Patients with Normal Spirometry

**DOI:** 10.3390/biomedicines10092129

**Published:** 2022-08-31

**Authors:** Tamas Nagy, Nora Melinda Toth, Erik Palmer, Lorinc Polivka, Balazs Csoma, Alexandra Nagy, Noémi Eszes, Krisztina Vincze, Enikő Bárczi, Anikó Bohács, Ádám Domonkos Tárnoki, Dávid László Tárnoki, György Nagy, Emese Kiss, Pál Maurovich-Horvát, Veronika Müller

**Affiliations:** 1Department of Pulmonology, Semmelweis University, 1083 Budapest, Hungary; 2Medical Imaging Centre, Semmelweis University, 1082 Budapest, Hungary; 3Department of Genetics, Cell- and Immunobiology, Semmelweis University, 1082 Budapest, Hungary; 4Department of Rheumatology and Clinical Immunology, Department of Internal Medicine and Oncology, Semmelweis University, 1083 Budapest, Hungary; 5Department of Clinical Immunology, Adult and Pediatric Rheumatology, National Institute of Locomotor Diseases and Disabilities, 1023 Budapest, Hungary; 6Department of Haematology and Internal Medicine, Semmelweis University, 1088 Budapest, Hungary

**Keywords:** systemic sclerosis, interstitial lung disease, cough, pulmonary hypertension, predictors of treatment response

## Abstract

Interstitial lung disease (ILD) is the leading cause of mortality in systemic sclerosis (SSc). Progressive pulmonary fibrosis (PPF) is defined as progression in 2 domains including clinical, radiological or lung-function parameters. Our aim was to assess predictors of functional decline in SSc-ILD patients and compare disease behavior to that in idiopathic pulmonary fibrosis (IPF) patients. Patients with normal forced vital capacity (FVC > 80% predicted; SSc-ILD: *n* = 31; IPF: *n* = 53) were followed for at least 1 year. Predictors of functional decline including clinical symptoms, comorbidities, lung-function values, high-resolution CT pattern, and treatment data were analyzed. SSc-ILD patents were significantly younger (59.8 ± 13.1) and more often women (93 %) than IPF patients. The median yearly FVC decline was similar in both groups (SSc-ILD = −67.5 and IPF = −65.3 mL/year). A total of 11 SSc-ILD patients met the PPF criteria for functional deterioration, presenting an FVC decline of −153.9 mL/year. Cough and pulmonary hypertension were significant prognostic factors for SSc-ILD functional progression. SSc-ILD patients with normal initial spirometry presenting with cough and PH are at higher risk for showing progressive functional decline.

## 1. Introduction

Systemic sclerosis (SSc) is a chronic connective tissue disease with multiple organ involvement [1]. Skin and internal organ manifestations are common; however, SSc-associated interstitial lung disease (SSc-ILD) is the leading cause of mortality in SSc [2]. Another significant cardiopulmonary complication accounting for additional high mortality is pulmonary hypertension (PH) [3,4]. The clinical course of this disease shows a high variability in progression [5].

Goh criteria are widely used to assess the severity of pulmonary involvement in SSc based on the extent of lung fibrosis on high-resolution computed tomography (HRCT) scan and forced vital capacity (FVC) % predicted value [6]. According to these criteria, SSc-ILD patients are classified into limited or extensive disease subgroups; however, functional decline in patients with normal spirometry, progression of symptoms, or HRCT is not included in the disease stratification [4]. The SSc-ILD Safety and Efficacy of Nintedanib in Systemic Sclerosis (SENSCIS) clinical trial confirmed that the annual rate of FVC decline in placebo-treated patients was on average −93.3 mL/year, and 46% of participants without specific treatment lost over 5% of FVC during a 1-year follow-up [7]. Furthermore, recent studies additionally suggested that SSc-ILD patients might lose lung function even in the physiological range and should be followed more closely to prevent functional deterioration [8].

Idiopathic pulmonary fibrosis (IPF) is a progressive fibrosing ILD of unknown origin and is associated with a poor outcome. IPF is characterized by a progressive functional decline that is mainly assessed using FVC and diffusing capacity of the lung for carbon monoxide (DL_CO_) [9]. FVC is currently the most extensively studied marker of disease progression in IPF, as this functional value is the only parameter accepted by regulatory bodies in predicting mortality [10].

Progressive pulmonary fibrosis (PPF) is defined by two parameters of the following three criteria: (1) worsening of respiratory symptoms; (2) physiological criteria of PFT decline (FVC ≥ 5%/year and/or DL_CO_ ≥ 10%/year); and (3) radiological signs of disease progression within a 1-year follow-up [11]. PPF might show similar clinical, lung functional, and radiological features over long term with IPF, the prototype of rapidly progressive ILD.

Our goal was to assess the PPF criteria on lung functional decline and possible clinical predictors of functional progression in SSc-ILD patients with normal spirometry and compare data with that of IPF patients to identify possible common factors defining functional progression.

## 2. Materials and Methods

### 2.1. Study Design and Parameters

Our study was a retrospective longitudinal observational study. All patients referred between February 2015 and January 2021 to the ILD multidisciplinary team (MDT) of the Department of Pulmonology, Semmelweis University, Budapest, Hungary, were screened. We enrolled subjects in our study who had been diagnosed with either SSc-ILD or IPF, had physiologic spirometric lung-function parameters (forced vital capacity (FVC) > 80% of predicted value), and had at least 12 months of follow-up data. SSc-ILD patients (classified with both J84 and M34 codes according to the 10th edition of the International Classification of Diseases (ICD10)) were diagnosed between January 2017 and July 2019. The diagnosis of SSc was established previously based on the American College of Rheumatology/European League Against Rheumatism Collaborative Initiative (EULAR-ACR) criteria [12] by immunology and rheumatology specialists; the disease-specific therapy was coordinated at immunological–rheumatological centers in central Hungary. Patients diagnosed with IPF between February 2015 and January 2021 according to the 2011 American Thoracic Society/European Respiratory Society (ATS/ERS) IPF guideline included in the EMPIRE registry were reviewed and all entered into the analysis if they met the inclusion criteria of having lung-function follow-up of at least one year with a baseline FVC > 80% predicted [10,13]. The study’s functional and follow-up criteria were met by 53 IPF and 31 SSc-ILD patients (Figure 1).

Baseline characteristics including smoking history, symptoms, detailed pulmonary function test (PFT) values (FVC, forced expiratory volume in 1st second (FEV_1_), total lung capacity (TLC), DL_CO_, carbon monoxide transfer coefficient (KL_CO_)). HRCT pattern and treatment data were analyzed and compared between the study populations. Arterial blood gas (ABG) analysis, body mass index (BMI), and 6-minute walk test (6MWT) results were examined. The gender, age, and physiology (GAP) index was calculated as a prognostic staging system [14]. At each follow-up visit, routine PFTs, ABGs, and the 6MWT were documented. Methods of the measurements (PFTs, ABGs, 6MWT, and HRCT) were described in detail in our previous study [8]. The median follow-up was 34 months and the functional decline of PPF was established throughout the follow-up according to the criteria of ≥5% FVC decline and/or ≥10% DL_CO_ decline within 1 year. A PPF diagnosis was not made because no additional second criterion of worsening symptoms and/or progression on HRCT was assessed [11].

### 2.2. Statistical Analysis

Data were analyzed using Graph Pad software (GraphPad Prism 5.0 Software, Inc., La Jolla, CA, USA) and SPSS v25 (IBM Corporation, Armonk, NY, USA). Continuous variables were expressed as the mean ± standard deviation (SD) or median (interquartile range (IQR)) and were compared using a *t*-test or a Mann–Whitney U-test according to the distribution of the variable. The test for normality was performed using the Kolmogorov–Smirnov test. Categorical variables are presented as percentages (%) expressed for the entire study population (all patients) or respective subgroups as indicated and were compared using a chi-squared test or two-tailed Fisher’s exact test. Multiple logistic regression analysis was used to assess predictors of possible functional progression including age (as a continuous variable), sex (male/female), smoking history (present/absent), cough (present/absent), PH (present/absent), baseline FVC, TLC, DL_CO_, KL_CO_ (all functional parameters in % predicted as continuous variables), and therapy (applied/none). A *p*-value < 0.05 was defined as statistically significant.

## 3. Results

The characteristics of the patients are summarized in Table 1. The mean age of the SSc-ILD patients was significantly lower than in IPF patients, with female predominance. IPF patients had been more frequently smokers and a higher proportion of patients qualified as overweight. A majority of the patients in both groups were in GAP stage I at baseline.

ILD-associated symptoms showed a significant difference between the two groups. More patients presented with respiratory symptoms such as dyspnea, cough, and crackles at the time of the diagnosis in the IPF group. On the other hand, the Raynaud phenomenon was only observed in the SSc-ILD group. No difference between groups was observed regarding the presence of PH and gastroesophageal reflux disease The Scl-70 antibody was present in 48% of SSc-ILD patients. The HRCT pattern was in line with actual guidelines and the literature, as usual interstitial pneumonia (UIP) or probable (p)UIP was the predominant pattern in IPF, while non-specific interstitial pneumonia (NSIP) predominated in SSc-ILD.

SSc therapy in 26 patients’ was conventional immunosuppressive therapy (ISU), while biological treatment was given in 9 cases, including 7 subjects with combined ISU and biological therapy. Antifibrotic agents (nintedanib or pirfenidone) were applied in 39 cases in the IPF group, while 14 patients did not receive IPF-specific treatment.

The functional parameters for the patients are summarized in Table 2. The values for FVC and FEV_1_ were similar in both groups, while TLC was significantly impaired and below the normal range in IPF patients. Baseline CO diffusion parameters showed a significant difference; DL_CO_ was decreased in the IPF group and SSc-ILD patients had lower KL_CO_ values. In the ABG, IPF patients had significantly lower pO_2_ values compared to SSc-ILD patients. The 6MWT distance was similar for both groups, representing a diminished functional capacity in SSc-ILD patients since normal percentile values are age-dependent [15,16]. Post-exercise desaturation in the 6MWT was present in both groups to a similar degree.

The annual median decline in FVC for SSc-ILD was −67.5 (−146.0 to −4.0) mL/year and −65.3 (−173.8 to −65.3) mL/year in the IPF group. During the follow-up period, 11 (35%) SSc-ILD patients met the criteria of PPF for functional decline: 7 patients presented with ≥5% FVC and 6 patients presented with ≥10% DL_CO_ predicted value decline, including 3 patients that met both criteria. In IPF patients, FVC and/or DL_CO_ decline, as defined for PFF, was present in 16 (30.2%) cases: 14 patients presented with ≥5% FVC and 7 patients presented with ≥10% DL_CO_ predicted value decline, including 5 patients meeting both criteria.

The annual median decline in FVC for the functionally progressive SSc-ILD subgroup was −153.9 (−278.3 to −121.4) mL/year, which was significantly higher compared to −26.2 (−75.4 to −1.6) mL/year in the stable/improved SSc-ILD subgroup (*p* = 0.017). In the functionally progressive IPF and stable/improved IPF subgroups, the respective data were −264.7 (−404.9 to −204.6) vs. −39.2 (−85.7 to +7.5) mL/year (*p* = 0.004). FVC decline for all patients, including functionally stable/improved and functionally progressive subgroups regarding IPF and SSc-ILD, are presented in Figure 2.

The functionally progressive IPF and SSc-ILD subgroups’ baseline characteristics and functional data are presented in Table 3. While in IPF, the functional progression was not associated with any difference in baseline characteristics or treatment, in SSc-ILD, cough and the presence of PH was significantly more common in the subgroup of patients showing functional decline over the one-year follow-up.

A multiple logistic regression did not show a significant baseline predictor of progression in the IPF group; however, in the SSc-ILD group, cough and PH were prognostic factors for functional progression (odds ratio: 36.2 (95% confidence interval: 1.8–711.9) and 36.4 (95% confidence interval: 1.1–1184.9), respectively). Most patients had a dry cough (SSc-ILD: *n* = 7; IPF: *n* = 17), and in the functionally progressive subgroups, a dry cough was predominant (SSc-ILD: 85.7% and IPF: 71.4%). The functional decline in individual patients according to treatment is presented in Figure 3.

## 4. Discussion

Our study confirmed that around one-third of SSc-ILD patients who had initial physiologic lung function parameters showed a decline in FVC and/or DL_CO_ over one year. While the disease course of SSc-ILD is heterogeneous, development of PPF might be a leading cause of death in this rare patient group. Although pulmonary functional decline is only one out of the three criteria defining PPF, PFT is an important non-invasive measurement that, in combination with symptoms, might promote clinicians to engage in early treatment interventions if PPF is confirmed [11].

According to our data, SSc patients with HRCT-confirmed ILD who are experiencing cough as the leading respiratory symptom might be at higher risk of developing PPF even with normal lung function results when presented at the respiratory specialist. Similarly, cough was also studied in a subgroup analysis of the SENSCIS trial and was found to be an important prognostic factor for functional decline and progression in SSc-ILD patients [17]. However, patients enrolled in the SENSCIS trial had less preserved lung function and patients with cough had an average decline of −95.6 mL/year, which was slightly higher than in our study. Importantly, nintedanib was less effective in SSc-ILD patients with a cough compared to patients without a cough. These two independent observations could mean that cough is an independent negative prognostic factor for functional decline. Cough was also found to be a marker of unfavorable therapeutic response in the Scleroderma Lung Study (SLS) II and was strongly correlated with lung-function decline [18,19,20]. It is important to note that the mean FVC decline in prospective average population cohort studies ranged in men between −47.2 and −78.4 mL/year, while in women it ranged between −14.1 and −65.6 mL/year, underlining functional loss in non-progressive patients within the expected range [21].

Additionally, our study confirmed the important prognostic role of PH for functional decline in SSc-ILD, which, to the best of our knowledge, has not been previously established in the literature. On the other hand, PH is a well-known severe complication of SSc, and several large investigations have confirmed its contribution to mortality [22,23,24]. However, our study was not powered to investigate PH as a risk of mortality due to the short observation period.

Surprisingly, we found that the 6MWT results did not differ between the SSc-ILD and IPF patients, although the SSc-ILD patients were significantly younger and therefore were expected to have better functional performance compared to older IPF patients. The underlying mechanisms might have included vascular manifestations such as PH and musculoskeletal conditions affecting exercise limitation [25,26,27].

IPF can be regarded as the prototype of rapidly progressive fibrotic lung disease. The common cellular mechanism in IPF and SSc-ILD is fibroblast activation and recruitment, as well as myofibroblast differentiation leading to interstitial extracellular matrix accumulation and lung fibrosis [9,15]. Our data confirmed that functional decline in our total IPF patient population was lower than in the nintedanib-treated population of the Safety and Efficacy of nintedanib at High Dose in IPF Patients (INPULSIS) trial (FVC decline of −65.3 vs. adjusted annual rate of change of −114.7 mL/year) [28]. Importantly, no difference was found in FVC decline between IPF patients receiving antifibrotics and the treatment-free subgroups. This highlighted that despite therapy, some patients might deteriorate even while receiving frontline antifibrotic therapy; therefore, further additional risk factors need to be identified. Notably, the original trials did not focus on patients with physiological lung function. The total IPF group (including untreated individuals) revealed a limited FVC decline; the stable/improved subgroup only showed a marginal loss in this parameter, while the progressive subgroup presented with a similar decline to that observed in patients on a placebo in clinical trials [28,29]. This real-world observation of IPF patients supported the beneficial effect of antifibrotic treatment, similar to our previous study on functionally advanced IPF cases [13].

Treatment of SSc-ILD is challenging. Our data confirmed better functional outcome in ISU and/or biological-therapy-treated SSc-ILD, emphasizing early and specific SSc treatment [8,30]. An SENSCIS subanalysis evaluated the effectiveness of nintedanib as an antifibrotic agent among SSc-ILD patients with or without baseline mycophenolate mofetil (MMF) treatment [7]. The inclusion criteria were: an FVC of at least 40% of the predicted value, a DL_CO_ of 30 to 89% of the predicted value, and a minimum of 10% fibrotic lung involvement on HRCT. At the end of the 52-week period, patients in the nintedanib + MMF group showed a significantly lower annual rate of decline in FVC than patients in the placebo + MMF group (−40.2 vs. −66.5 mL/year), and an even higher functional decline was present in the absence of ISU treatment (SSc-ILD on placebo: −119.3 mL/year). PPF SSc-ILD patients on the ISU therapy MMF with the addition of nintedanib had the highest advantage in lung-function preservation [7,31].

Treatment of PPF is still not well established; however, antifibrotic therapy might be an option in selected cases [11]. Nintedanib proved to be effective in reducing the annual rate of FVC decline in different fibrosing ILD patients in the Efficacy and Safety of Nintedanib in Patients with Progressive Fibrosing ILDs (INBUILD) trial and in the INPULSIS 1-2 IPF trials [28,32].

Optimal timing and dosing of ISU and/or biological therapy in SSc is of the utmost importance. Higher awareness of therapy initiation is needed in patients with normal lung function presenting with cough and in the presence of PH, and close pulmonary follow-up of these patients is suggested [33]. 

The limitations of our study were the retrospective single-center design and the low number of patients, which did not allow for a better stratification according to clinical features or treatment. Further prospective studies are needed to establish new markers of progression and to develop guidelines for the optimal timing of treatment introduction, with the adequate therapies being a combination of ISU and/or biological therapy and/or antifibrotics in the case of PPF in this special subgroup of SSc-ILD patients [34].

## 5. Conclusions

Progression in ILDs has an unfavorable effect on several clinical outcomes. Regular measurements of FVC are one of the most important parameters accepted for monitoring functional decline among IPF and SSc-ILD subjects. Patient-reported symptoms such as cough (especially dry cough) and the presence of PH as a lung-related comorbidity should be taken into consideration in connection with the disease progression in SSc-ILD patients. Defining high-risk patients for PPF is of utmost importance because timely and optimal introduction of ISU and/or biological and possibly antifibrotic agents might prevent disease deterioration. However, the timing and combination of treatments require further research in SSc-ILD and other PPF. Close monitoring and regular follow-ups are required in patients with normal lung function, especially in patients presenting with cough and PH. 

## Figures and Tables

**Figure 1 biomedicines-10-02129-f001:**
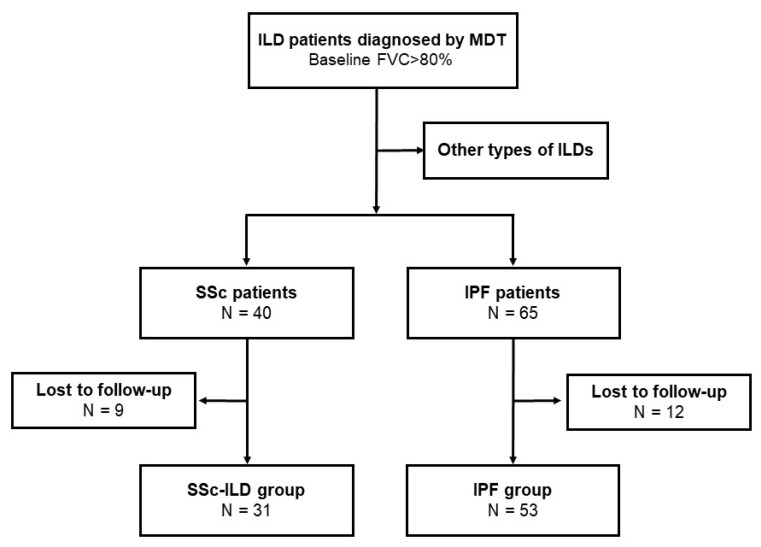
Patient selection for analysis. FVC, forced vital capacity; ILD, interstitial lung disease; IPF, idiopathic pulmonary fibrosis; MDT, multidisciplinary team; SSc-ILD, systemic-sclerosis-associated interstitial lung disease.

**Figure 2 biomedicines-10-02129-f002:**
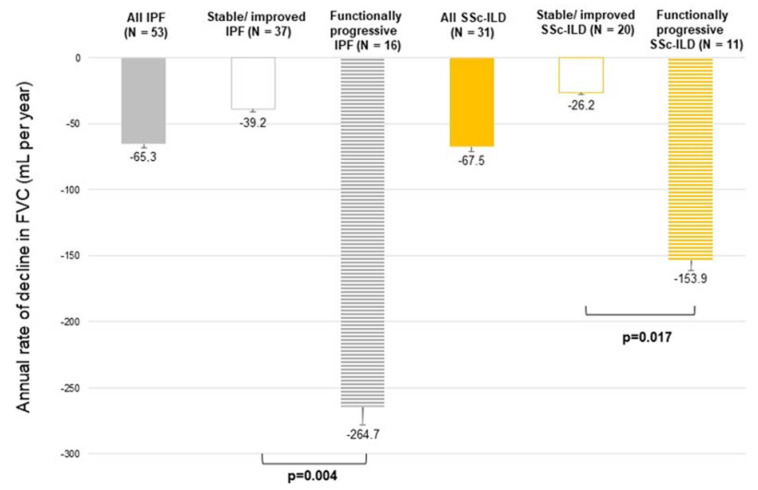
Annual rate of decline in FVC (mL per year) in IPF and SSc-ILD patients. FVC, forced vital capacity; IPF, idiopathic pulmonary fibrosis; SSc-ILD, systemic-sclerosis-associated interstitial lung disease.

**Figure 3 biomedicines-10-02129-f003:**
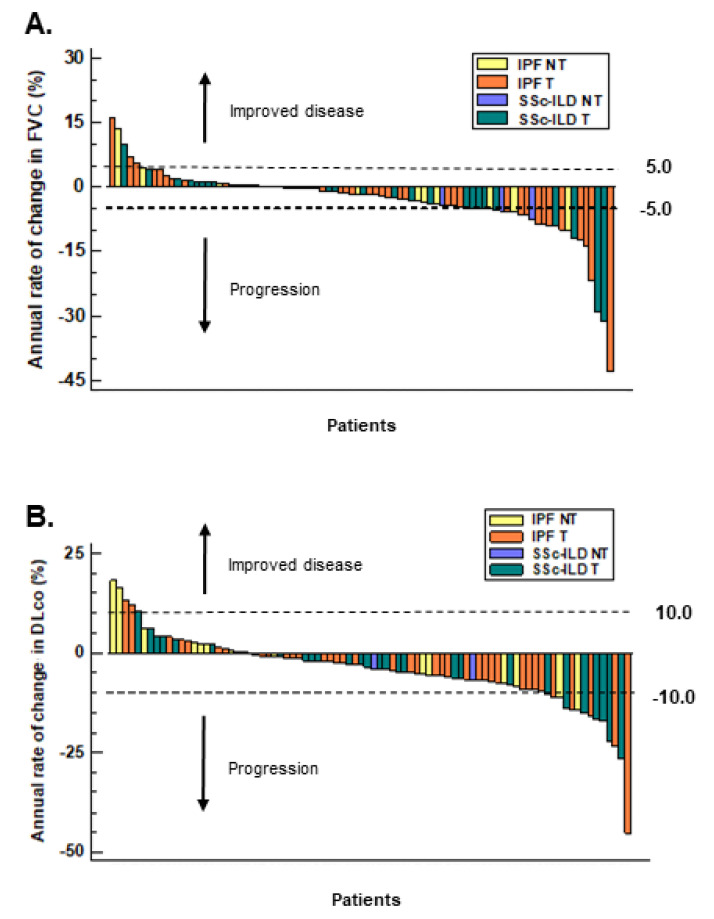
Annual changes in FVC (**A**) and DL_CO_ (**B**) % of the predicted value in all IPF and SSc-ILD patients according to specific treatment groups. DL_CO_, diffusing capacity for carbon monoxide; FVC, forced vital capacity; IPF, idiopathic pulmonary fibrosis; NT, no treatment; SSc-ILD, systemic-sclerosis-associated interstitial lung disease; T, treatment.

**Table 1 biomedicines-10-02129-t001:** Patient characteristics.

Characteristics	IPF (*n* = 53)	SSc-ILD (*n* = 31)	*p*-Value
Age (years)	68.9 ± 8.5	59.8 ± 13.1	**0.001**
Sex (male:female)	28:25	2:29	**<0.001**
Smoking history			
Ever smoker	33 (62.3)	7 (22.6)	**0.001**
Non-smoker	20 (37.7)	24 (77.4)	
BMI (kg/m^2^)	27.7 ± 4.4	25.2 ± 4.4	**0.006**
Overweight (25.0–29.9 kg/m^2^)	21 (39.6%)	5 (16.2%)	**0.025**
Signs and symptoms			
Dyspnea	52 (98.1)	11 (35.5)	**<0.001**
Cough	26 (49.1)	9 (29.0)	0.108
Finger clubbing	12 (22.6)	0	NA
Crackles	47 (88.7)	12 (38.7)	**<0.001**
Raynaud phenomenon	0	23 (74.2)	NA
GAP score			
Stage I	50 (94.3)	31 (100.0)	NA
Stage II	3 (5.7)	0	NA
Stage III	0	0	NA
Specific comorbidities			
PH	7 (13.2)	5 (16.1)	0.712
GERD	6 (11.3)	6 (19.4)	0.345
HRCT pattern			
UIP/pUIP	26/25 (96.2)	0/3 (9.7)	**<0.001**
NSIP	0	26 (83.9)	NA
Other	2 (3.8)	2 (6.5)	0.578
Therapy			
Nintedanib	39 (73.6)	0	NA
Pirfenidone	8 (15.0)	0	NA
ISU	0	26 (83.9)	NA
Biological treatment	0	9 (29.0)	NA
None	14 (26.4)	3 (9.7)	NA

Data are presented as *n* (%) or mean ± SD. BMI, body mass index; GAP, Gender–Age–Physiology index; GERD, gastro-esophageal reflux disease; HRCT, high-resolution computed tomography; IPF, idiopathic pulmonary fibrosis; NA, not assessed; NSIP, non-specific interstitial pneumonia; PH, pulmonary hypertension; pUIP, probable usual interstitial pneumonia; SSc-ILD, systemic-sclerosis-associated interstitial lung disease; UIP, usual interstitial pneumonia; ISU, immunosuppressive therapy. GAP index: stage I = 0–3 points; stage II = 4–5 points; stage III = 6–8 points. Statistically significant values are highlighted in bold.

**Table 2 biomedicines-10-02129-t002:** Baseline lung function, ABG, and 6MWT functional parameters.

Values	IPF (*n* = 53)	SSc-ILD (*n* = 31)	*p*-Value
Lung-function parameters			
FVC (mL)	3035.7 ± 836.2	2725.5 ± 655.6	0.080
FVC (% pred)	96.4 ± 13.9	98.7 ± 12.2	0.263
FEV_1_ (mL)	2488.5 ± 696.4	2301.6 ± 569.2	0.209
FEV_1_ (% pred)	98.6 ± 16.2	99.7 ± 13.3	0.748
FEV_1_/FVC (%)	82.6 ± 8.2	84.5 ± 5.2	0.329
TLC (mL)	4648.9 ± 1358.4	4263.9 ± 823.3	0.157
TLC (% pred)	79.5 ± 14.4	88.4 ± 15.4	**0.022**
Diffusion parameters			
DL_CO_ (mmol/min/kPa)	5.9 ± 1.8	6.4 ± 1.6	0.201
DL_CO_ (% pred)	74.1 ± 17.6	83.7 ± 18.3	**0.020**
KL_CO_ (mmol/min/kPa/L)	1.3 ± 0.3	1.4 ± 0.3	**0.042**
KL_CO_ (% pred)	88.8 ± 24.2	71.2 ± 16.4	**<0.001**
ABGs			
pH	7.4 ±0.0	7.4 ±0.0	0.655
pCO_2_ (mmHg)	37.7 ± 5.5	37.1 ± 2.3	0.119
pO_2_ (mmHg)	67.8 ± 11.0	78.6 ± 8.6	**<0.001**
6MWT			
Distance (m)	454.4 ± 103.1	449.3 ± 70.8	0.502
Initial SpO_2_ (%)	95.3 ± 2.9	94.9 ± 3.0	0.463
Final SpO_2_ (%)	88.8 ± 9.0	89.2 ± 10.8	0.407
Initial HR (1/min)	81.0 ± 13.9	84.4 ± 14.3	0.425
Final HR (1/min)	111.2 ± 19.7	106.5 ± 20.2	0.443
Initial Borg score (0–10)	0 (0–0)	0 (0–0)	0.885
Final Borg score (0–10)	2 (0–4)	1.5 (1–3)	0.924

Data are presented as mean ± SD, median (IQR). 6MWT, 6-minute walk test; ABGs, arterialized capillary blood gases; DL_CO_, diffusing capacity of the lungs for carbon monoxide; FEV_1_, forced expiratory volume in 1 s; FVC, forced vital capacity; HR, heart rate; IPF, idiopathic pulmonary fibrosis; KL_CO_, transfer coefficient of the lung for carbon monoxide; pCO_2_; partial pressure of carbon dioxide; pO_2_, partial pressure of oxygen; SpO_2_, oxygen saturation; SSc-ILD, systemic-sclerosis-associated interstitial lung disease; TLC, total lung capacity. Statistically significant values are highlighted in bold.

**Table 3 biomedicines-10-02129-t003:** Patient characteristics; HRCT pattern; treatment; and baseline lung function, ABG, and 6MWT functional parameters of the IPF and SSc-ILD subgroups.

Values	Functionally Stable/Improved IPF (*n* = 37)	Functionally Progressive IPF (*n* = 16)	Functionally Stable/Improved SSc-ILD (*n* = 20)	Functionally Progressive SSc-ILD (*n* = 11)
Age (years)	67.6 ± 9.1	71.8 ± 6.5	59.8 ± 13.1	59.7 ± 12.6
Sex (male:female)	20:17	8:8	1:19	1:10
Smoking history				
Ever smoker	23 (62.2)	10 (62.5)	3 (15.0)	4 (36.4)
Non-smoker	14 (37.8)	6 (37.5)	17 (85.0)	7 (63.6)
BMI (kg/m^2^)	28.3 ± 4.4	26.8 ± 4.3	25.2 ± 4.4	23.7 ± 4.1
Overweight (25.0–29.9 kg/m^2^)	15 (40.5%)	6 (37.5%)	5 (45.5%)	0
Signs and symptoms				
Dyspnea	37 (100)	15 (93.8)	7 (35.0)	4 (36.4)
Cough	19 (51.4)	7 (43.8)	**2 (10.0) ***	**7 (63.6) ***
Finger clubbing	6 (16.2)	6 (37.5)	0	0
Crackles	33 (89.2)	14 (87.5)	7 (35.0)	5 (45.5)
Raynaud phenomenon	0	0	16 (80.0)	7 (63.6)
GAP score				
Stage I	35 (94.6)	15 (93.8)	20 (100.0)	11 (100.0)
Stage II	2 (5.4)	1 (6.2)	0	0
Stage III	0	0	0	0
Specific comorbidities				
PH	5 (13.5)	2 (13.3)	**1 (5.0) ^#^**	**4 (36.4) ^#^**
GERD	5 (13.5)	1 (6.2)	3 (15.0)	3 (27.3)
HRCT pattern				
UIP/pUIP	16/21 (100)	10/4 (87.5)	0/3 (15.0)	0
NSIP	0	0	16 (80.0)	10 (90.9)
Other	0	2 (12.5)	1 (5.0)	1 (9.1)
Therapy				
Nintedanib	27 (73.0)	12 (75.0)	0	0
Pirfenidone ^&^	3 (8.1)	5 (31.2)	0	0
ISU	0	0	18 (90.0)	8 (72.7)
Biological treatment	0	0	7 (35.0)	2 (18.2)
None	10 (27.0)	4 (25.0)	1 (5.0)	2 (18.2)
Lung-function parameters				
FVC (mL)	3057.0 ± 840.8	2986.3 ± 850.7	2770.5 ± 681.2	2642.7 ± 631.4
FVC (% pred)	95.5 ± 12.8	98.4 ± 16.5	98.9 ± 13.7	98.4 ± 9.7
FEV_1_ (mL)	2497.6 ± 707.6	2467.5 ± 692.0	2354.5 ± 609.4	2200.9 ± 510.8
FEV_1_ (% pred)	96.8 ± 15.0	102.7 ± 18.5	100.0 ± 14.7	98.6 ± 11.3
FEV_1_/FVC (%)	82.0 ± 8.7	82.9 ± 5.2	84.8 ± 4.5	83.6 ± 6.4
TLC (mL)	4683.5 ± 1459.1	4568.8 ± 1130.2	4343.0 ± 884.2	4199.1 ± 635.5
TLC (% pred)	79.7 ± 16.1	78.8 ± 9.5	89.7 ± 17.5	88.1 ± 10.7
Diffusion parameters				
DL_CO_ (mmol/min/kPa)	6.1 ± 1.8	5.3 ± 1.6	6.3 ± 1.5	6.6 ± 1.6
DL_CO_ (% pred)	76.6 ± 18.0	68.3 ± 15.5	82.5 ± 18.4	88.6 ± 14.5
KL_CO_ (mmol/min/kPa/L)	1.3 ± 0.4	1.2 ± 0.2	1.4 ± 0.3	1.5 ± 0.3
KL_CO_ (% pred)	89.2 ± 23.6	87.8 ± 26.3	70.3 ± 16.3	75.6 ± 12.9
ABGs				
pH	7.4 ±0.0	7.4 ±0.0	7.4 ±0.0	7.4 ±0.0
pCO_2_ (mmHg)	38.5 ± 2.6	35.3 ± 10.0	37.7 ± 1.9	36.0 ± 2.8
pO_2_ (mmHg)	69.7 ± 10.2	62.2 ± 11.8	77.0 ± 6.4	81.4 ± 11.7
6MWT				
Distance (m)	459.7 ± 102.5	442.5 ± 107.1	454.7 ± 68.8	438.4 ± 81.8
Initial SpO_2_ (%)	95.6 ± 2.3	94.6 ± 3.8	95.6 ± 1.9	93.6 ± 4.4
Final SpO_2_ (%)	89.4 ± 8.9	87.4 ± 9.3	93.6 ± 5.3	82.0 ± 14.0
Initial HR (1/min)	80.4 ± 12.9	82.3 ± 16.4	85.6 ± 15.6	82.4 ± 13.2
Final HR (1/min)	109.7 ± 19.7	114.7 ± 19.7	106.0 ± 16.7	107.2 ± 27.2
Initial Borg score (0–10)	0 (0–0)	0.5 (0–0)	0 (0–0)	0 (0–1)
Final Borg score (0–10)	1.5 (0–4)	2.8 (0–4)	2 (1–3)	1 (0–3)

Data are presented as *n* (%) or mean ± SD, median (IQR). 6MWT, 6-minute walk test; ABGs, arterialized capillary blood gases; BMI, body mass index; DL_CO_, diffusing capacity of the lungs for carbon monoxide; FEV_1_, forced expiratory volume in 1 s; FVC, forced vital capacity; GAP, Gender–Age–Physiology index; GERD, gastro-esophageal reflux disease; HR, heart rate; HRCT, high-resolution computed tomography; IPF, idiopathic pulmonary fibrosis; ISU, immunosuppressive therapy; KL_CO_, transfer coefficient of the lung for carbon monoxide; NSIP, non-specific interstitial pneumonia; pCO_2_; partial pressure of carbon dioxide; PH, pulmonary hypertension; pO2, partial pressure of oxygen; pUIP, probable usual interstitial pneumonia; SpO2, oxygen saturation; SSc-ILD, systemic-sclerosis-associated interstitial lung disease; TLC, total lung capacity; UIP, usual interstitial pneumonia;. GAP index: stage I = 0–3 points; stage II = 4–5 points; stage III = 6–8 points. Statistically significant values are highlighted in bold: * *p* = 0.002; # *p* = 0.023. ^&^ All treatment was included: a nintedanib-to-pirfenidone or pirfenidone-to-nintedanib change in therapy resulted in the higher number of treated vs. untreated IPF patients.

## Data Availability

The original contributions presented in the study are included in the article; further inquiries can be directed to the corresponding author.

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
