# Peer review of "Clinical Predictors of Lung-Function Decline in Systemic-Sclerosis-Associated Interstitial Lung Disease Patients with Normal Spirometry"

_biomedicines, 2022, doi:10.3390/biomedicines10092129_

Round 1
Reviewer 1 Report
In SSc ILD the authors applied PPF criteria on lung functional decline and possible clinical predictors of progression in those with normal spirometry: the data were comapred to IPF patients. the low number of patients and the significant heterogeneity of SSc patients does not really allow to draw any conclusion with the present data. In fact, patients are not stratified according to clinical features.
Author Response
Dear Reviewer 1,
enclosed please find the revised version of our manuscript entitled Clinical predictors of lung function decline in systemic sclerosis-associated interstitial lung disease patients with normal spirometry to publish in Biomedicines.
We are grateful for the effort of the Reviewers and the Editorial Office, and we hope that our responses will meet the satisfaction of the journal.
As requested, please find our point-by-point responses to the comments below.
Please see the attachment.
Yours sincerely,
Authors

Reviewer 2 Report
The authors have studied lung function in systemic sclerosis with the aim to uncover predictors of worsening lung function among patients with normal PFTs, including comparison to subjects with idiopathic pulmonary fibrosis. I have the following comments -
1. The Method section state that subjects were entered into the analyses if inclusion criteria were met, but this criteria are not given. Are these the inclusion criteria for the EMPIRE Registry? I think the criteria needed to be given.
2. A large number of comparisons are made between the groups. For instance, as shown in Table 1 and 3. I do not see definite apriori hypotheses. So, correction of the p values for multiple comparisons may be in order. But perhaps need for correction is mitigated by the regression analysis.
3. The confirmation of cough as a prognostic indicator is an important finding. Is there anything else to know about cough in scleroderma for which the authors might have data? Timing of the cough - nocturnal? Sputum production versus not?
Author Response
Dear Reviewer 2,
enclosed please find the revised version of our manuscript entitled Clinical predictors of lung function decline in systemic sclerosis-associated interstitial lung disease patients with normal spirometry to publish in Biomedicines.
We are grateful for the effort of the Reviewers and the Editorial Office, and we hope that our responses will meet the satisfaction of the journal.
As requested, please find our point-by-point responses to the comments below.
Please see the attachment.
Yours sincerely,
Authors

Reviewer 3 Report
The primary objective of this review was to evaluate PPF criteria for decreased lung function and possible clinical predictors of functional progression in SSc-ILD patients with normal lung function and to compare disease behavior with patients with idiopathic pulmonary fibrosis (IPF). Patients with normal cardiopulmonary capacity (FVC >80% predicted; SSc-ILD: N=31; IPF: N=53) were monitored for at least 1 year, and the analysis included clinical symptoms, comorbidities, pulmonary function values, high-resolution CT patterns, treatment data, and other predictors of functional decline, found that patients with SSc-ILD had normal initial vital capacity, and patients with cough and PH had a higher risk of progressive decline in function.
Some comments are as follows
1. The authors compared patients with SSc-ILD to patients with IPF. Are there any similarities between IPF and SSc-related ILD in terms of the pathophysiology leading to pulmonary fibrosis?
2. It is mentioned in the Intordution that "SSc-ILD patients are classified into limited or extensive disease groups.", does this statistical analysis generalize these two types of patients? Because the extent of the violation of the two organs is not the same.
3. If there are combined therapy recommendations in Table 1 and 3 Therapy, they can be expressed separately, so that the treatment methods can be more intuitively understood from the table.
4. FVC depends on their age, gender, etc. Compared with IPF, SSc-ILD patients are mostly female. Does it affect the difference?
5. The X-axis characters in Figure 3 are too small to see clearly.
Author Response
Dear Reviewer 3,
enclosed please find the revised version of our manuscript entitled Clinical predictors of lung function decline in systemic sclerosis-associated interstitial lung disease patients with normal spirometry to publish in Biomedicines.
We are grateful for the effort of the Reviewers and the Editorial Office, and we hope that our responses will meet the satisfaction of the journal.
As requested, please find our point-by-point responses to the comments below.
Please see the attachment.
Yours sincerely,
Authors

Round 2
Reviewer 1 Report
the authors accept the comment on the low number of patients as well as the heterogeneity of their cohort. Regretfully, both these points do not allow a sound evalution even in a real life setting.
Reviewer 3 Report
The author has fully responded to the previous question, and I have no further suggestions.